# Tug-of-war between actomyosin-driven antagonistic forces determines the positioning symmetry in cell-sized confinement

Ryota Sakamoto [1,9], Masatoshi Tanabe[2,9], Tetsuya Hiraiwa [3,4], Kazuya Suzuki[2,5], Shin'ichi Ishiwata [2], Yusuke T. Maeda [1] & Makito Miyazaki [6,7,8 ✉]

Symmetric or asymmetric positioning of intracellular structures including the nucleus and mitotic spindle steers various biological processes such as cell migration, division, and embryogenesis. In typical animal cells, both a sparse actomyosin meshwork in the cytoplasm and a dense actomyosin cortex underneath the cell membrane participate in the intracellular positioning. However, it remains unclear how these coexisting actomyosin structures regulate the positioning symmetry. To reveal the potential mechanism, we construct an in vitro model composed of cytoplasmic extracts and nucleus-like clusters confined in droplets. Here we find that periodic centripetal actomyosin waves contract from the droplet boundary push clusters to the center in large droplets, while network percolation of bulk actomyosin pulls clusters to the edge in small droplets. An active gel model quantitatively reproduces molecular perturbation experiments, which reveals that the tug-of-war between two distinct actomyosin networks with different maturation time-scales determines the positioning symmetry.

[1] Department of Physics, Graduate School of Science,  Kyushu University, 744 Motooka, Nishi-ku, Fukuoka 819-0395, Japan. [2] Department of Physics, Faculty of Science and Engineering,  Waseda University, 3-4-1 Okubo, Shinjuku-ku, Tokyo 169-8555, Japan. [3] Department of Physics, The University of Tokyo, 7-3-1 Hongo, Bunkyo-ku, Tokyo 113-0033, Japan. [4] Mechanobiology Institute, National University of Singapore, Singapore 117411, Singapore. [5] Central Research Laboratory, Hamamatsu Photonics K.K., 5000 Hirakuchi, Hamakita-ku, Hamamatsu, Shizuoka 434-8601, Japan. [6] Hakubi Center for Advanced Research, Kyoto University, Yoshida-honmachi, Sakyo-ku, Kyoto 606-8501, Japan. [7] Department of Physics, Graduate School of Science, Kyoto University, Kitashirakawa Oiwake-cho, Sakyo-ku, Kyoto 606-8502, Japan. [8] Institut Curie, PSL Research University, CNRS, UMR 144, F-75005 Paris, France. [9]These authors contributed equally: Ryota Sakamoto, Masatoshi Tanabe. ✉email: miyazaki.makito.3s@kyoto-u.ac.jp

Regulation of intracellular positioning is important for various cellular functions. Asymmetric positioning of the nucleus is always coupled with the directed motility in motile cells[1]. During embryogenesis, symmetric nuclear positioning is correlated with successful maturation[2,3], while failure of the asymmetric mitotic spindle positioning yields an abnormal chromosomal number, causing birth defects disease[4].

Both symmetric and asymmetric positioning of the nucleus and spindle are regulated by the actin cytoskeleton. Hierarchical assembly of nano-scale actin and myosin molecules in cytoplasm generates micrometer-long to submillimeter-long mechanical interactions, which orchestrates macroscopic geometry sensing. For example, symmetry breaking of bulk actomyosin flow followed by asymmetric positioning of the nucleus initiates cell migration in fish keratocytes[5]. Active diffusion of actin-coated vesicles in cytoplasm driven by myosin V generates a pressure gradient, which targets the nucleus at the center in mouse oocytes[6]. After the nuclear centration, the nucleus transforms into a spindle apparatus and migrates to the oocyte surface, where a dense actin meshwork formed around one of the two spindle poles initially closer to the cell membrane is involved in the asymmetric positioning[7,8]. Recent studies suggest that not only the bulk actomyosin but also actomyosin localized beneath the cell membrane is involved in the geometry sensing. For example, the cortex thickening induced by Arp2/3-mediated F-actin assembly guides asymmetric spindle positioning in mouse oocytes[9].

Importantly, two distinct actomyosin structures often coexist in the same system, i.e., bulk actomyosin networks and actomyosin cortex beneath the cell membrane, and actomyosin-regulated positioning of the nucleus and spindle is two-state, i.e., either at the center or close to the membrane boundary. These facts raise a fundamental question how the two-state geometry sensing (center or edge) is functioned by the two actomyosin structures located in spatially distinct places (bulk and surface). However, the inherent complexity of the cell, including coexistence of several distinct actomyosin networks and regulatory signals from the nucleus and spindles, makes it difficult to dissect the direct contribution of each actomyosin structure on the positioning of intracellular structures. Hence, a comprehensive understanding of the actomyosin-regulated geometry sensing mechanism is still missing.

In vitro reconstitution approaches aiming to understand physical mechanisms of the self-organization of active cytoskeleton have been extensively explored[10–20]. Field et al. cestablished a method to prepare actin-intact cytoplasmic extracts of Xenopus eggs and observed periodic gelation contraction[21]. Spatial confinement of the extracts into droplets mimicking the cell boundary displays various actomyosin dynamics observed in living cells, such as symmetry breaking of the actin cortex[16,19] and spontaneous F-actin retrograde flow[13,22], providing insights into physical mechanisms of the cytoskeleton self-organization.

In this study, we investigate how the two sorts of actomyosin structures in the bulk and surface can mechanically control the positioning of intracellular structures, by employing the in vitro model composed of Xenopus egg extracts and a single nucleus-like spherical body confined in a droplet. This model not only allows us to modulate the surface–actomyosin interactions and the bulk actomyosin network properties, but also allows us to change the system size to modulate the surface to volume ratio, by which we can quantitatively evaluate each contribution of the surface and bulk actomyosin on the spatial positioning. In addition, since the model is not expected to contain any nucleus-specific and mitotic spindle-specific regulatory signals[9,23], purely physical contributions of actomyosin could be identified. Here, we found that the positioning symmetry was steered by a tug-of-war between two antagonistic forces generated by actomyosin waves contracting toward the droplet center and percolated actomyosin networks connecting the nucleus-like spherical body and the droplet boundary. A theoretical model based on the active gel theory quantitatively reproduced the size-dependent two-state positioning, and predicted modulation of the transition droplet diameter was demonstrated by molecular perturbations of actin crosslinkers and lengths of actin filaments. These findings will help us understand the regulatory mechanism of intracellular symmetry, that is, ruled by a synergy between actomyosin-driven active mechanics and geometric constraints imposed by the cell boundary.

## Results

**Cluster formation and periodic actomyosin wave generation.** We employed metaphase Xenopus egg extracts as a model of the cytoplasm, and mimicked the cell boundary by encapsulating the extracts into water-in-oil droplets surrounded by a monolayer of natural phospholipids (Fig. 1a and b). Droplets were prepared in accordance with a standard emulsification procedure by gently mixing the extracts and lipid–oil mixture[17,18,24,25]. Consequently, poly-dispersed droplets from few microns to ~300 μm in diameter were obtained. Immediately, these droplets were sandwiched between polydimethyl-siloxane (PDMS)-coated glass slides to render the droplets in a quasi-two-dimensional configuration for simplicity. Actin filaments were visualized by using tetra-methylrhodamine (TMR)-labeled LifeAct, which specifically binds to actin filaments but not to actin monomers. This small peptide of concentration of 1 μM has only minor effects on actin dynamics and mechanical properties of actomyosin networks[26–28]. Furthermore, we added nocodazole, a microtubule polymerization inhibitor, to eliminate the effects of microtubules on the actin cytoskeleton.

Extract-in-oil droplets were prepared on ice. The temperature was then increased to 20 °C to initiate actin polymerization and actin dynamics were observed using an epi-fluorescence microscope (Fig. 1c–e). Actin filaments were immediately polymerized throughout the droplets, forming an F-actin network. Approximately 10 s after increasing the temperature, the entire network started to contract (Fig. 1c, Supplementary Movie 1). The network contraction accumulated cytoplasmic materials including organelles (Fig. 1f), forming a single nucleus-like spherical body (hereinafter called a "cluster"). The droplet boundary disrupted the force balance at the interface, hence, the actomyosin network had contracted[13].

After cluster formation, new actin networks were formed near the droplet boundary (Supplementary Fig. 1), and they self-organized into periodic waves propagating from the boundary to the center (Fig. 1d, e, and g, Supplementary Movie 1). When the actin wave approached the cluster, the wave fused with the cluster. The periodic actin waves persisted more than 90 min (Supplementary Movie 2).

Periodic contractile waves have been observed in mm-scale bulk extracts, and based on the observation, a gelation-contraction mechanism has been proposed[21]. We confirmed that the extracts used in the present study show similar gelation-contraction behaviors in the bulk (Supplementary Fig. 2, Supplementary Movie 3) and the contractility was regulated in a cell cycle as demonstrated by the fact that the interphase extracts did not form clear clusters and generate periodic actomyosin waves (Supplementary Fig. 3), both of which were consistent with the previous study[21]. However, under the cell-sized confinement, we observed clear ring-shaped actomyosin waves (Fig. 1d and e), rather than homogeneous contraction of the actomyosin gel (Supplementary Fig. 2). In mm-scale extracts,

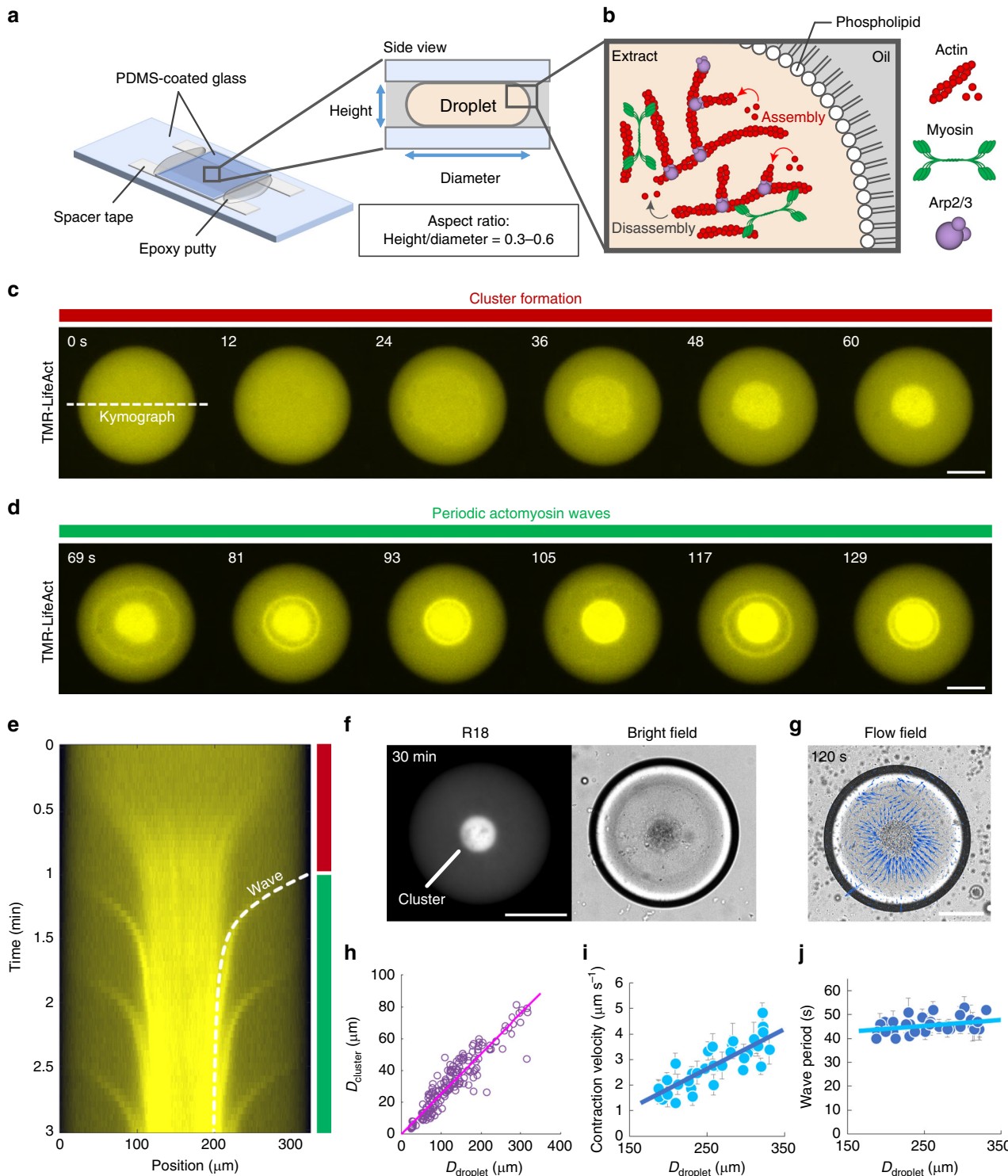

both the F-actin density at the edge of the wave and the F-actin density in the bulk space inside of the edge increased during the contraction (Supplementary Fig. 2d–f), showing that the actomyosin network was contracted almost homogeneously. In contrast, under a cell-sized confinement, the F-actin density on the ring increased during its contraction, while the F-actin density in the bulk space inside of the ring remained nearly constant (Supplementary Fig. 4), indicating that the ring region contracted faster than the bulk region inside of the ring, thereby only the density of the ring significantly increased during contraction.

Indeed, when the ring was cut by UV laser ablation, it immediately opened up in a few seconds (Supplementary Fig. 5, Supplementary Movies 4 and 5), showing that there had existed a strong lateral contractile force in the ring. These observations strongly suggest that the ring predominantly generates the contractile force during wave propagation, qualitatively different from the contraction behavior observed in mm-scale bulk extracts. Although the mechanism causing those differences is unclear, we focus on actomyosin behaviors in a cell-sized confined space in the present study.

**Fig. 1 Cell-sized confinement induces cluster formation and periodic actomyosin waves. a** Schematic illustration of the experimental setup. The extract-in-oil droplets were confined in a quasi-two-dimensional space between two polydimethyl siloxane (PDMS)-coated glass slides. The aspect ratio of the height to diameter was fixed at 0.3–0.6. **b** Magnified view of the droplet boundary. The droplet was surrounded by a single layer of natural phospholipids to mimic the cell boundary. Actin filaments are nucleated by the Arp2/3 complex. Myosin induces actin network contraction. **c–e** Time-lapse images of F-actin dynamics in the extract-in-oil droplet, showing **c** initial contraction of the F-actin network, followed by **d** periodic wave propagation, and **e** the kymograph (Supplementary Movie 1). The broken line shows the theoretical model of the actomyosin wave (Supplementary Note 1). Actin filaments were visualized by tetramethylrhodamine (TMR)-LifeAct. The droplet temperature was elevated from 0 to 20 °C at 0 s to initiate actin polymerization. Periodic actomyosin waves persisted for more than 90 min (Supplementary Movie 2). We performed >10 independent experiments and confirmed the repeatability. **f** Fluorescence and the bright-field images of organelles stained with Octadecyl Rhodamine B chloride (R18). Organelles were accumulated by the initial contraction of the actomyosin network, forming a single nucleus-like spherical body (called a "cluster"). **g** Velocity field of the actomyosin wave visualized by particle imaging velocimetry. The same sample as **c** and **d** was analyzed. **h** Relationship between the cluster diameter $D_{cluster}$ and the droplet diameter $D_{droplet}$ ($n = 176$). The plot was fitted by $D_{cluster} = 0.25D_{droplet}$ ($R^2 = 0.86$; R-squared value for linear regression). We performed two independent experiments. **i, j** Initial contraction velocity $v$ and period of the wave $T$ displayed along the droplet diameter $D_{droplet}$ ($n = 35$). The plots were fitted by $v = 1.6 \times 10^{-2}D_{droplet} - 1.3$ ($R^2 = 0.69$) and $T = 2.5 \times 10^{-2}D_{droplet} + 39$ ($R^2 = 0.12$), respectively. Error bars represent standard deviations from the mean velocities and mean periods averaged over three successive waves and wave periods, respectively. We performed two independent experiments. Scale bars, 100 μm.

To identify which molecules were involved in wave generation and contraction, we treated the extracts with inhibitors and examined the effects. Compared to control experiments (Supplementary Fig. 6a, Supplementary Movie 6), partial inhibition of the Arp2/3 complex by CK666 extended the wave period (Supplementary Fig. 6b, Supplementary Fig. 7a and b, Supplementary Movie 6), indicating that actin polymerization nucleated by the Arp2/3 complex (Fig. 1b) was involved in the wave generation. In addition, enhanced myosin activity by calyculin A increased the wave velocity (Supplementary Fig. 6c, Supplementary Fig. 7c and d, Supplementary Movie 6). On the contrary, partial inhibition of myosin activity by Y27632 decelerated the velocity (Supplementary Fig. 6d, Supplementary Fig. 7c and d, Supplementary Movie 6). These results show that wave propagation was regulated by myosin. Furthermore, the waves disappeared by addition of an actin polymerization inhibitor cytochalasin D (Supplementary Fig. 8a, Supplementary Movie 7), while inhibition of actin depolymerization by phalloidin facilitated network shrinkage, but waves did not persist (Supplementary Fig. 8b, Supplementary Movie 8). Thus, continuous turnover of F-actin was also essential for periodic wave generation.

The cluster size was linearly correlated with droplet size (Fig. 1h), implying that the amount of cytoplasmic material accumulated by network contraction was proportional to the droplet volume. Here, we fixed the aspect ratio of chamber height $h$ to droplet diameter $D_{droplet}$ between $0.3 \leq h/D_{droplet} \leq 0.6$ in the following experiments to maintain a quasi-two-dimensional condition (Supplementary Movie 9) and not to interrupt the motion of clusters according to droplet size. We varied the droplet size from 16 to 330 μm in diameter, and found that the velocity of actomyosin waves was linearly correlated with droplet size (Fig. 1i), while the period of actomyosin waves had only a weak positive correlation with droplet size (Fig. 1j).

**Cluster position becomes off-centered in small droplets**. It has been reported that actin gels self-organized in extract-in-oil droplets are occasionally localized at the droplet boundary[13,19]. In the present study, we found that the positioning of nucleus-like clusters strongly depended on droplet size (Fig. 2a). Despite the radial symmetry of cylindrical droplets, the geometric center was no longer a stable point in small droplets. We characterized how far the cluster was apart from the droplet center, namely an extent of the positioning symmetry, by introducing a scale-free polarity parameter, the DC-ratio (Fig. 2b), defined as the distance between the droplet center and centroid of the cluster $d$ divided by the droplet radius $R_{droplet}$, i.e., DC-ratio = $d/R_{droplet}$.

After cluster formation, the clusters either rested near their initial position or began moving away from it. In some droplets, clusters moved from the droplet boundary to the center (Fig. 2c–e, Supplementary Movie 10), while others exhibited outward movement (Fig. 2f–h, Supplementary Movie 11). Thereafter, they found a stable position within ~30 min. Large droplets ($D_{droplet} \gtrsim 100$ μm) showed symmetric cluster positioning in the center (Fig. 2a, top). By contrast, most clusters were positioned at the edge in small droplets ($D_{droplet} \lesssim 100$ μm) (Fig. 2a, bottom). A series of experiments for differently sized droplets from 16 to 330 μm in diameter revealed that the cluster position displayed a sharp transition from the center to edge with a reduction in droplet size (Fig. 2i–k).

**Actin-membrane interaction controls the cluster positioning**. What determines the cluster position? By utilizing advantages of the reconstituted system, we first tested whether the surface property of the droplet boundary affected the cluster positioning (Fig. 3a). When we replaced natural phospholipids with polyethylene glycol (PEG)-based passive surfactants to reduce adhesion of proteins on the droplet boundary (Fig. 3a, top), the clusters favored to locate in the droplet center (Fig. 3b, top). On the other hand, when we promoted actin polymerization on the droplet surface by activating Arp2/3 on the membrane via the VCA domain of WASP (the constitutively active form) (Fig. 3a, bottom), the clusters favored to locate at the edge (Fig. 3b, bottom). The period of actomyosin waves was not much affected in both cases (Supplementary Figs. 9 and 10). Since it was expected that physical interaction of the bulk actomyosin network with the droplet boundary was reduced by surface passivation and increased by Arp2/3 localization, these results suggested that the off-centering was steered by the physical interaction between the bulk actomyosin network and the droplet boundary.

**Actomyosin bridges pull the cluster to the droplet boundary**. During observation, we found by chance the transient oscillatory motion of the cluster, in which the cluster periodically moved back and forth along the radial direction (Fig. 4a and b, Supplementary Movie 12). This motion suggests coexistence of two antagonistic forces acting on the cluster: a centripetal force and an outward radial force. Reflecting observations that the surface passivation promoted centering of the cluster (Fig. 3), but neither the wave propagation behavior nor the period was critically changed (Supplementary Figs. 9 and 10), we speculated that the actomyosin wave was a source of the centripetal force. We confirmed that the cluster positioned between the edge and the center

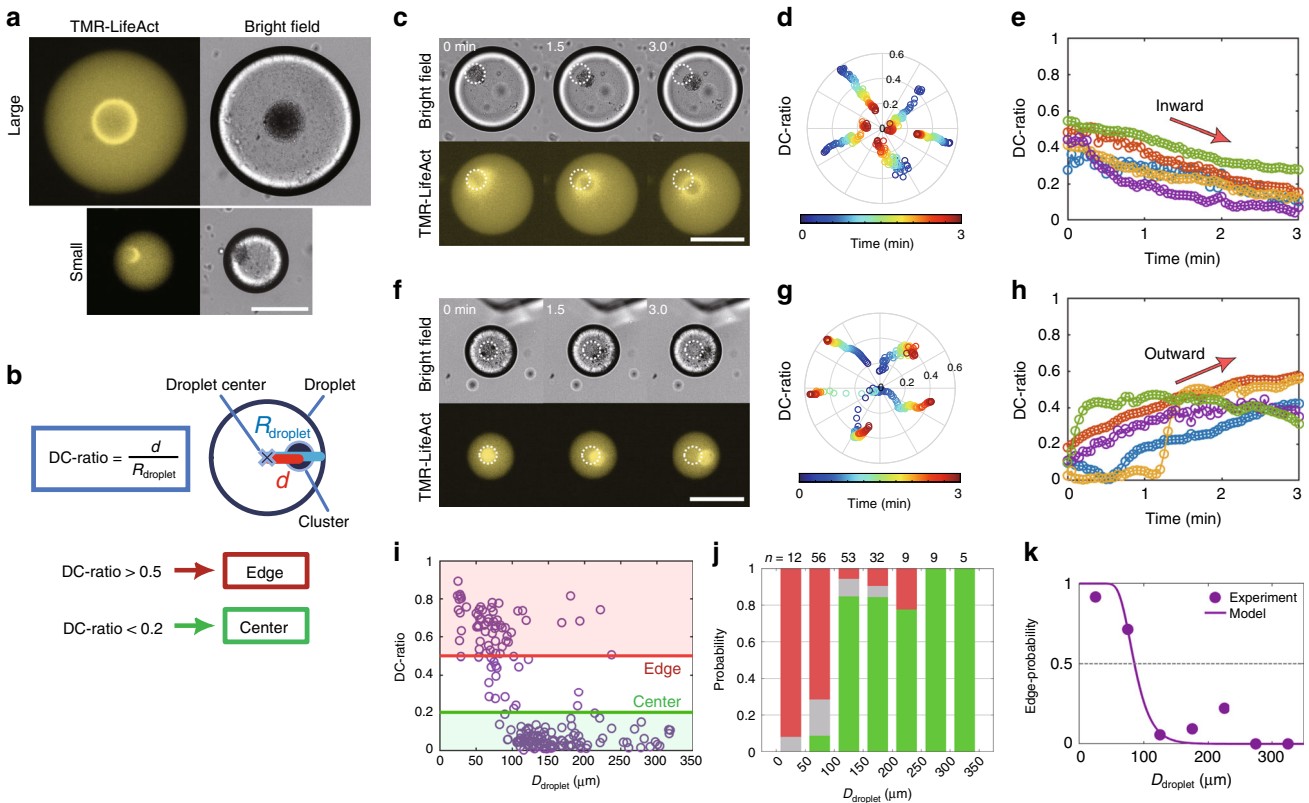

**Fig. 2 Cluster position becomes off-centered as the droplet size decreases. a** Typical examples of the droplet size dependence. Stable position of the cluster became off-centered in small droplets. Images were acquired 1 h after encapsulation. **b** Quantification of the cluster positioning symmetry. The polarity parameter DC-ratio is defined as the ratio of $d$ to $R_{droplet}$, where $d$ is the distance from the droplet center to the cluster centroid, and $R_{droplet}$ is the radius of the droplet. DC-ratio is classified into two distinct regimes: larger than 0.5 is the "edge" and smaller than 0.2 is in the "center". **c–h** Typical examples of the **c–e** inwardly and **f–h** outwardly directed motion of clusters after their formation (Supplementary Movies 10 and 11). The time point at which clusters started to move (typically ~1 min after the cluster formation) is defined as 0 min. The clusters either rested near their initial positions or moved away from the initial positions. Only the clusters that moved are shown. Different colors in **e** and **h** indicate different clusters. **i–k** Droplet size dependence of cluster position. **i** Raw data of DC-ratio showing two-state positioning. The stable position depends on the droplet diameter. We analyzed bright-field images acquired more than 1 h after encapsulation. We performed two independent experiments ($n = 176$ droplets). **j** Histogram of DC-ratio for each 50 μm bins summarized from **i**. DC-ratio > 0.5, 0.5 ≥ DC-ratio ≥ 0.2, and 0.2 > DC-ratio are colored in red, gray, and green, respectively. **k** Edge-positioned probability calculated from **j**. Filled circles and the solid curve represent experimental values and model fitting of Eq. (2), respectively. Experimental data were fitted by Eq. (2) using $L = 6.1$ μm and $\tau = 0.46$ s. Transition diameter $D_c = 85$ μm at which edge-probability becomes 0.5 was estimated from the fitting curve. Scale bars, 100 μm.

moved toward the center when the waves approached the cluster (Supplementary Fig. 11). For the outward radial force, bright field observation revealed that several bundles were connected with the cluster and the droplet boundary when the cluster showed oscillatory motion (Fig. 4a, yellow arrowheads). To confirm that these bridges are consisting of actin filaments, we first tried to visualize the bridge formation process by a confocal microscope. It turned out that it was difficult to obtain clear images of individual bridges because of low contrast (Supplementary Movie 13). Hence, we performed laser ablation experiments with simultaneous observation of bulk actin networks: we cut the networks between the droplet boundary and the cluster that was approaching to the edge, and observed the response (Fig. 4c, Supplementary Movie 14). Just after the laser ablation, the cluster began to move back to the droplet center. Then, after several minutes, the cluster began to move to the edge again. In the same time, F-actin density between the cluster and the nearest droplet boundary increased, which appeared to reassemble the bridge (Fig. 4c, white arrow). This observation suggests that the actomyosin bridges pull the cluster to the edge.

Based on these observations, we propose a tug-of-war model (Fig. 4d). We hypothesize that actomyosin waves propagating

from the droplet boundary to the center periodically push the cluster toward the droplet center (Fig. 4d, top), whereas actomyosin bridges pull the cluster toward the boundary (Fig. 4d, bottom). Considering that the formation of actomyosin bridges between the cluster and the droplet boundary is a stochastic process, the expected time for its formation $\tau_p$ has a strong size dependence because abundant crosslinked points are required to transmit the contractile force along the long distance (Fig. 4d, bottom zoom-in views). By contrast, we experimentally verified that the period of actomyosin waves $T$ has only a weak dependence on droplet size (Fig. 1j). Therefore, with a reduction in droplet size, $\tau_p$ should become smaller than $T$ at a certain droplet diameter, and thus the positioning symmetry might be broken.

**Actin-associated proteins shift the transition diameter.** To verify the tug-of-war model, we investigated whether the transition diameter from centered to edged positioning was altered with changes in the stoichiometry of actin-associated components in the bulk actomyosin networks. We defined the transition diameter $D_c$ at which the edge-positioned probability was $p = 0.5$. To

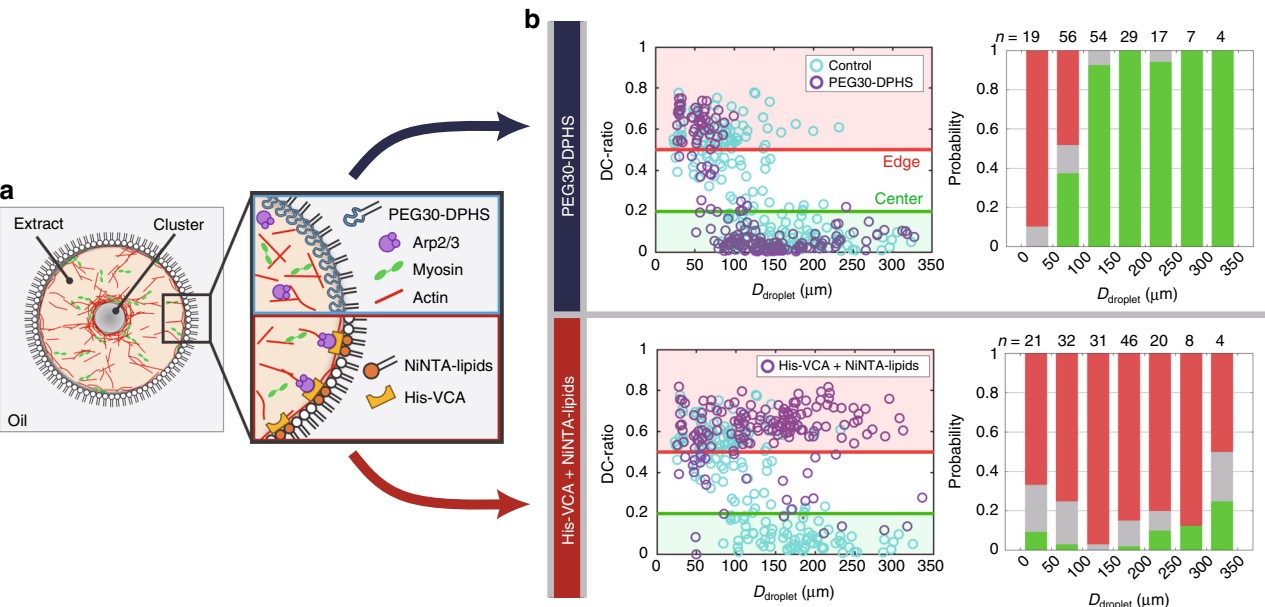

**Fig. 3 Modulations of the surface property alters the cluster positioning.** The interaction between the droplet boundary and actomyosin networks in the bulk space was modulated by changing the surface property. **a** Schematic illustration showing alternations of the surface property of droplets. (Top) PEG30-DPHS, a polyethylene glycol (PEG)-based surfactant, will decrease physical interactions of bulk actomyosin networks with the droplet boundary. (Bottom) The VCA domain of WASP conjugated with a histidine-tag was anchored to the droplet boundary via Ni-NTA-conjugated lipids. VCA recruits Arp2/3 and activates its actin nucleation activity. Thus, it was expected that interactions of bulk actomyosin networks with the droplet boundary increased. **b** The edge-positioned probabilities were compared between different surface properties. Cyan circles show the control data (egg PC). (Top) Passivated droplet surface promoted the cluster centering, whereas (bottom) activated droplet surface promoted the edge positioning. Kolmogorov–Smirnov test was applied to the scatter plots of DC-ratio between $D_{droplet} = 50\ \mu m – 150\ \mu m$ (around the transition point in the control experiment). The distributions were significantly different from the control in both conditions ($p < 0.001$). We performed two independent experiments.

finely evaluate this $D_c$, we fitted the analytical expression for the edge probability, derived from the tug-of-war model later (see "Methods" section for details), to a set of data.

Addition of the actin crosslinker $\alpha$-actinin increased the transition diameter from $D_c = 85\ \mu m$ (control, Fig. 2k) to $D_c = 115\ \mu m$ (Fig. 5, top, Supplementary Movie 15). In contrast, the period of actomyosin waves was $T \simeq 45\ s$, almost same as the control $T \simeq 46\ s$ (Figs. 1j and 5, top). These results indicate that the probability of actomyosin bridge formation between the cluster and the droplet boundary was enhanced, resulting in a shorter time-scale to contract the cluster towards the edge. Furthermore, we altered the length distribution of actin filaments in the bulk space of droplets. On the one hand, addition of the actin-severing protein gelsolin (constitutively active mutant) shifted the transition diameter from $D_c = 85\ \mu m$ (control) to $D_c = 60\ \mu m$ (Fig. 5, middle, Supplementary Movie 16), indicating that shorter actin filaments led to less efficient actomyosin bridge formation, and thus the cluster was mostly positioned at the center by the actomyosin waves. On the other hand, addition of formin mDia2 (constitutively active mutant), which nucleates actin polymerization and assembles long unbranched actin filaments, increased the transition diameter $D_c = 138\ \mu m$ (Fig. 5, bottom, Supplementary Movie 17), indicating longer-range force transmission by the actomyosin bridges. In both cases, the periods of actomyosin waves $T \simeq 41\ s$ (gelsolin) and $T \simeq 41\ s$ (mDia2) changed little from the control $T \simeq 46\ s$ (Figs. 1j and 5, middle and bottom). We also measured the lengths of individual F-actin polymerized in the extracts by a fluorescence microscope and confirmed that the length was shortened by gelsolin or elongated by mDia2 (Supplementary Fig. 12). Collectively, the network connectivity and F-actin length regulated by actin-associated proteins can regulate cluster positioning, and the present results are consistent with the prediction, supporting the tug-of-war model.

**Theoretical modeling of the two-state cluster positioning.** We next constructed an approximated but quantitative theory of the tug-of-war model. Bridge formation has its characteristic maturation time of $\tau_p$ to steer the off-centered cluster positioning, while inwardly traveling actomyosin waves push the cluster back to the geometric center at every period of $T$. It is expected that the shorter time-scale dominates cluster positioning, and if $\tau_p$ and $T$ have different droplet-size dependences, cluster positioning may show size-dependent sharp transition at the crossover point (Fig. 6a).

**Step 1: The bridge formation probability.** We first consider the formation process of actomyosin bridges. The process could be understood by a framework of the network percolation that has been extensively applied to various transport problems, such as fluid flow through random porous media[29]. The percolation theory describes the extent of network connectivity[30] and can predict the probability of the emergence of an inter-connected path through the entire system enabling the transmission of signals over a long distance[31]. Regarding actomyosin networks, myosin contractile forces can be transmitted over a long distance through actin filaments when an adequate number of crosslinking sites is occupied by crosslinkers, which corresponds to the force percolation transition[32–34].

We derive the percolation probability that the total number of $N$ actin-crosslinking sites in one dimension are inter-connected during a time interval $T$[30]. Actomyosin bridges are formed when the total number of $N$ sites are occupied by crosslinkers (Fig. 6b). In the simplest case, each crosslinking site takes "on" or "off" state for every turnover time of crosslinkers $\tau$ with probability of 1/2, assuming that the binding and unbinding are in equilibrium. Thus, the probability that all $N$ sites are simultaneously occupied

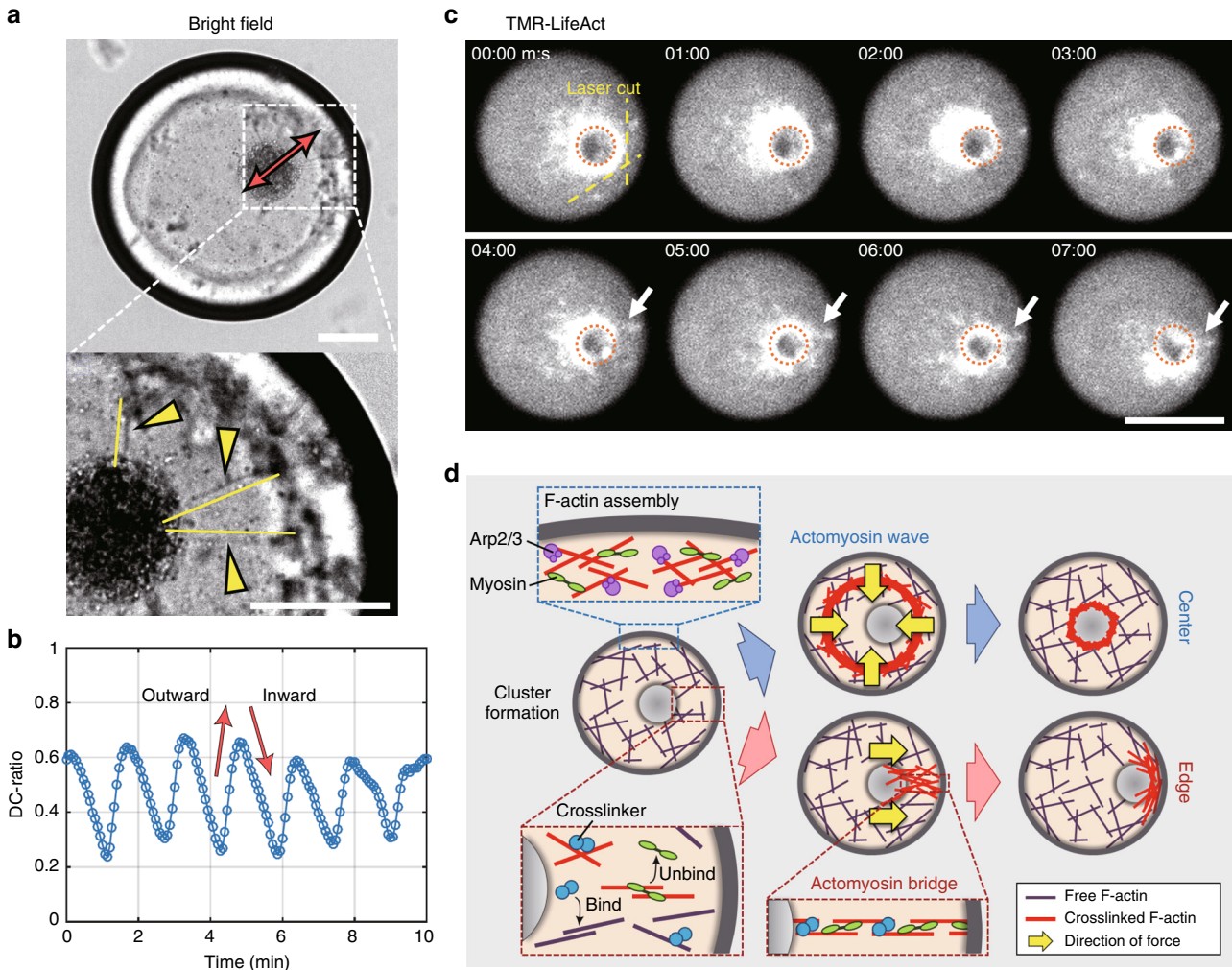

**Fig. 4 Actomyosin bridges pull the cluster toward the droplet boundary. a**, **b** Oscillatory motion of the cluster (Supplementary Movie 12). **a** Snapshot images of the droplet showing the oscillatory motion. (Top) The direction of motion of the cluster is indicated by the red double-sided arrow. (Bottom) The bridges connecting the cluster and the boundary are indicated by yellow arrow heads. **b** Time course of the cluster motion shown in **a**. **c** Actomyosin bridges formed between the cluster and the droplet boundary were cut by a UV pulsed laser. After the laser ablation, the cluster began to move back to the droplet center. After 4 min has passed, the cluster began to move to the edge again. In the same time, actomyosin bridges reassembled (white arrow). The original position of the cluster is indicated by the red broken circle (Supplementary Movie 14). **d** The tug-of-war model. (Top) Actomyosin waves push the cluster to the droplet center in every wave period. A magnified view showing F-actin polymerization nucleated by the Arp2/3 complex. (Bottom) Between periodic actomyosin waves, if actomyosin bridges are formed, network contraction pulls the cluster to the edge. A magnified view showing the dynamics of bulk actomyosin between periodic actomyosin waves. If inter-connected actomyosin bridges are formed between the cluster and the droplet boundary, the cluster is contracted toward the edge. Scale bars, 50 μm.

by crosslinkers at each time step is $(1/2)^N$. Therefore, we obtain the probability that all $N$ crosslinking sites are occupied at least one time during the time interval $T$ as

$$p(N) = 1 - \left[ 1 - \left( \frac{1}{2} \right)^N \right]^{T/\tau},\tag{1}$$

where $\tau = 1/(k_{off} + C_0 k_{on})$ is a characteristic turnover time of crosslinkers[35], $C_0$ denotes concentration of crosslinkers, and $k_{on}$ and $k_{off}$ denote binding and unbinding rates of crosslinkers on actin filaments, respectively. Given that the typical length of F-actin is $L$ and the droplet radius is $R$, the number of crosslinking sites between the droplet boundary and the center-positioned cluster is $N \simeq R/L$.

**Step 2: The wave period.** We next derive the period of actomyosin waves $T$ as a function of the droplet diameter $D = 2R$. We

consider the active gel dynamics with growth of an F-actin network (Fig. 6c). The actin ring can induce contractile stress after time $t_0$, in which F-actin is polymerized enough long to have contact points with one another (Fig. 6c, (i)). Thereafter, myosin and crosslinkers associate with the ring (Fig. 6c, (ii)), thereby the ring starts to contract (Fig. 6c, (iii)). Subsequently, Arp2/3 complexes remained outside of the ring nucleates new F-actin, and a new ring is assembled. Considering these sequential processes, the contraction period $T$ of the ring is given simply by $t_0$. In addition, the propagation profile of the actomyosin wave (Fig. 1e, broken line) and the size dependence of the contraction velocity (Fig. 1i) are also reproduced by the framework of active gel theory (Supplementary Fig. 13, Supplementary Note 1). Note that, if the droplet boundary has non-negligible effects through actin–membrane interactions, there is a possibility that the contractile force of myosin is not enough to break the interaction between the ring and the droplet boundary just after the

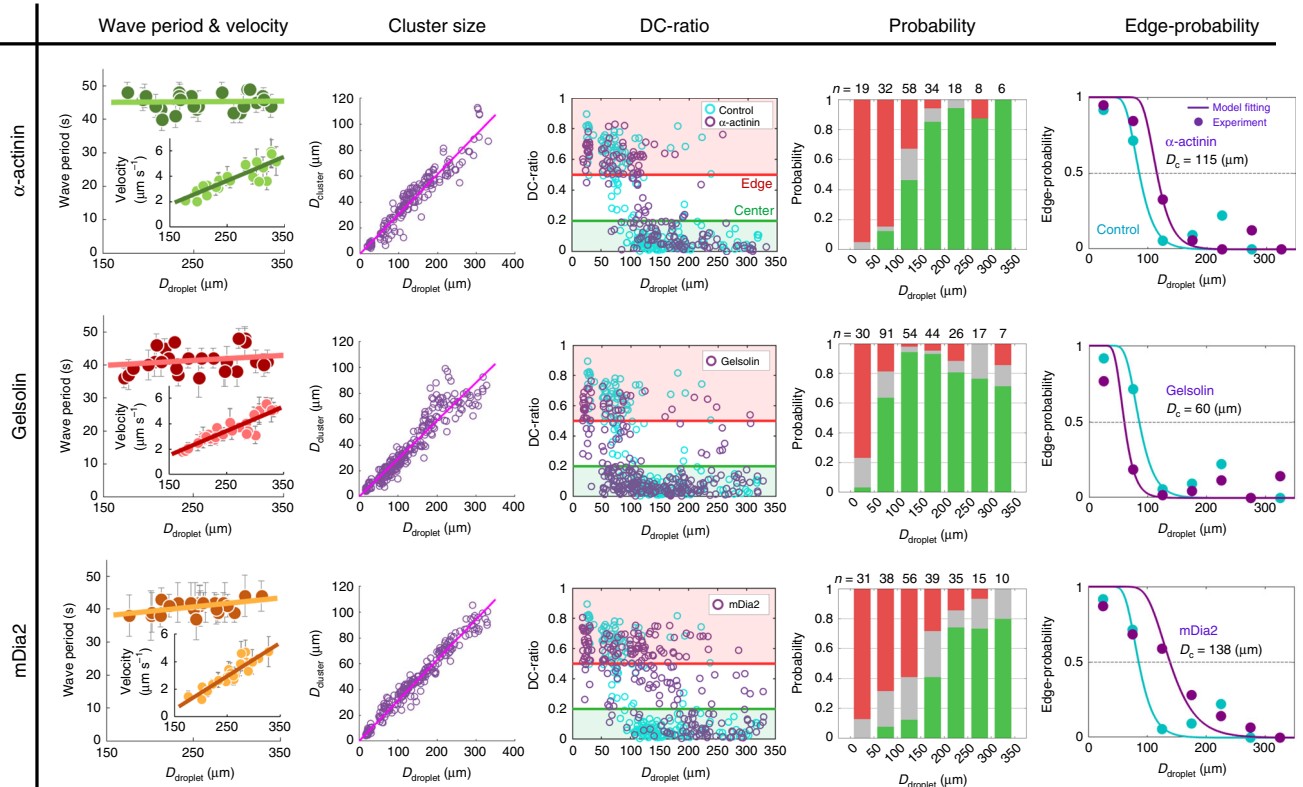

**Fig. 5 Modulation of the network connectivity and F-actin length alter the cluster positioning.** The connectivity and F-actin length of actomyosin networks in the bulk space were modulated by adding actin-associated proteins. Error bars displayed in the wave period and velocity represent standard deviations from the mean periods and mean velocity averaged over three successive wave periods and waves, respectively. Cyan circles in the DC-ratio show the control data. Cyan circles and curves in the Edge-probability show the control data and the model fitting, respectively. (*Top*) Addition of the actin crosslinker, $\alpha$-actinin (250 nM). The wave period $T$ and initial contraction velocity $v$ (inset) were fitted by $T = 1.8 \times 10^{-3} D_{droplet} + 45$ ($R^2 < 0.01$) and $v = 1.9 \times 10^{-2} D_{droplet} - 1.1$ ($R^2 = 0.76$), respectively ($n = 25$) (Supplementary Movie 15). Cluster diameter was fitted by $D_{cluster} = 0.31 D_{droplet}$ ($n = 175$, $R^2 = 0.91$). Edge-positioned probability was fitted by Eq. (2) using $L = 6.1\,\mu m$ and $\tau = 0.09\,s$. Transition diameter $D_c = 115\,\mu m$. (Middle) Addition of the actin severing protein, gelsolin (300 nM). The wave period $T$ and initial contraction velocity $v$ were fitted by $T = 1.5 \times 10^{-2} D_{droplet} + 38$ ($R^2 = 0.04$) and $v = 1.9 \times 10^{-2} D_{droplet} - 1.4$ ($R^2 = 0.81$), respectively ($n = 27$) (Supplementary Movie 16). The cluster diameter was fitted by $D_{cluster} = 0.29 D_{droplet}$ ($n = 269$, $R^2 = 0.89$). Edge-positioned probability was fitted by Eq. (2) using $L = 4.3\,\mu m$ and $\tau = 0.46\,s$. Transition diameter $D_c = 60\,\mu m$. (Bottom) Addition of actin polymerization activator, mDia2 (30 nM). The wave period $T$ and initial contraction velocity $v$ were fitted by $T = 2.7 \times 10^{-2} D_{droplet} + 34$ ($R^2 = 0.26$) and $v = 2.4 \times 10^{-2} D_{droplet} - 3.1$ ($R^2 = 0.83$), respectively ($n = 27$) (Supplementary Movie 17). The cluster diameter was fitted by $D_{cluster} = 0.31 D_{droplet}$ ($n = 224$, $R^2 = 0.96$). Edge-positioned probability was fitted by Eq. (2) using $L = 10\,\mu m$ and $\tau = 0.46\,s$. Transition diameter $D_c = 138\,\mu m$. We confirmed that addition of actin-associated proteins had little influence on the cluster size distribution compared to the control experiments (Fig. 1h). Kolmogorov–Smirnov test was applied to the scatter plots of DC-ratio between $D_{droplet} = 50\,\mu m$ – $150\,\mu m$ (around the transition point in the control experiment). The distributions were significantly different from the control in all conditions ($p < 0.01$). We performed two independent experiments.

maturation. In this case, our model predicts a weak size dependence of the period of actomyosin waves $T(D)$. Such a weak size dependence was observed in the experiments (Fig. 1j, Supplementary Note 2).

**Step 3: The edge-positioning probability.** We finally calculate the probability of symmetry breaking, i.e., the probability that the actomyosin network percolates between the cluster and the droplet wall during the wave period $T(D)$ (Supplementary Note 3). We assume that (i) the cluster is transported to the droplet center by actomyosin waves every period of $T(D)$, and (ii) the cluster is transported to the edge if actomyosin in the bulk space forms a bridge between the cluster and the droplet boundary, and the outward movement is stopped when the cluster collides with the subsequent actomyosin wave. We approximately assume that the cluster immediately moves to the edge if percolation occurs during the period $T(D)$; otherwise it is positioned at the center (two-state model). By using $N \simeq R/L = D/(2L)$ in the actomyosin bridge formation probability (Eq. (1)), we obtain the edge-

positioning probability as a function of the droplet diameter $D$:

$$p(D) = 1 - \left[ 1 - \left( \frac{1}{2} \right)^{D/(2L)} \right]^{T(D)/\tau}. \tag{2}$$

**Tug-of-war model reproduces the positioning transition.** An analytical expression of the tug-of-war model (Eq. (2)) (Fig. 7, solid lines) quantitatively reproduces the experimentally observed size-dependent transition (Fig. 2k solid line, Fig. 5, right, solid lines). Numerical simulations considering the cluster motion coupled with periodic actomyosin waves and stochastic network percolation were also performed, which reproduce the size-dependent transition as well (Fig. 7, markers, Supplementary Note 4, Supplementary Fig. 14, Supplementary Movies 18 and 19).

We use Eq. (2) to estimate physical quantities and the transition diameter $D_c$ at which $p(D_c) = 0.5$. Equation (2) is fitted to the results of control experiments with the following two parameters: the contour length of actin filaments $L$ and the turnover rate of crosslinkers $\tau$, and estimated $L = 6.1\,\mu m$, $\tau = 0.46\,s$, and

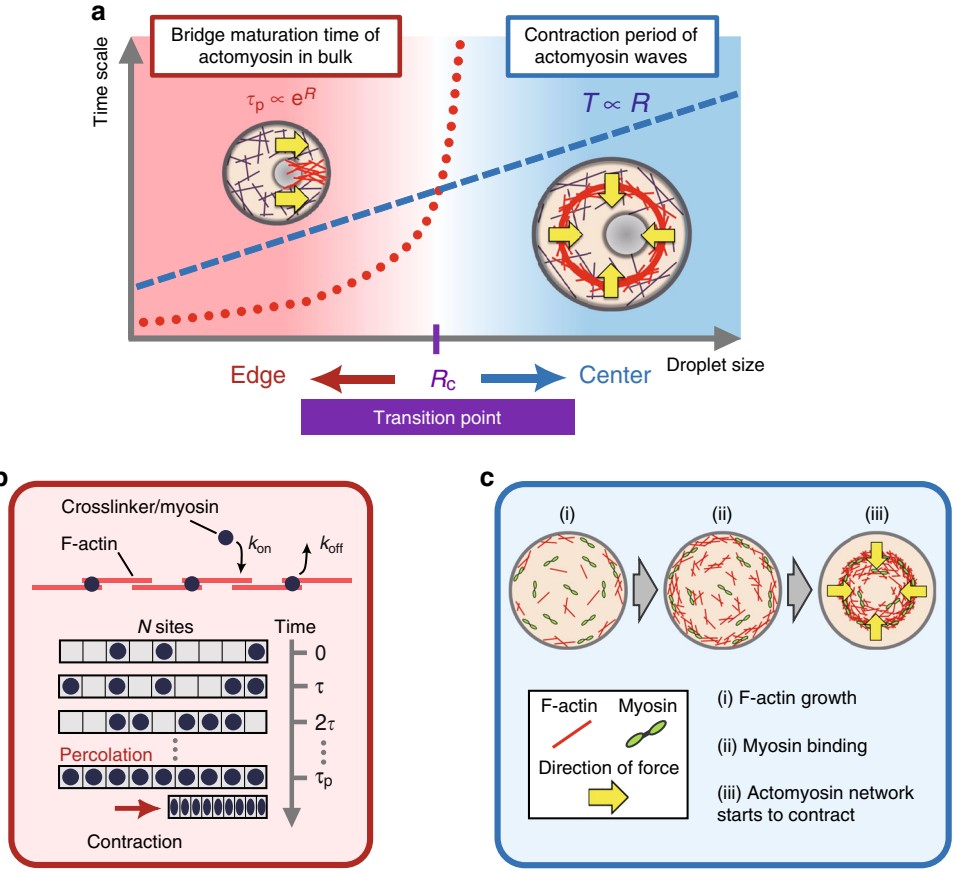

**Fig. 6 Theoretical modeling of the two-state cluster positioning. a** The proposed mechanism underlying droplet size-dependent two-state cluster positioning. The red dotted curve and blue dashed line represent the bridge maturation time $\tau_p$ and the wave period $T$, respectively. Actomyosin waves transport the cluster toward the droplet center every period of $T$, while stochastically formed actomyosin bridges (mean maturation time $\tau_p$) transport the cluster towards the droplet boundary. Actomyosin bridges are stochastically formed in the time period $T$ between actomyosin waves. If bridges are formed, the network contraction pulls the cluster toward the droplet boundary until the subsequent wave collides with the cluster. Since the characteristic time-scales of two antagonistic forces have different size dependences, the transition radius $R_c$ is determined by one unique crossover point. The resultant stable position is determined by the matured network faster than the other. **b** The percolation model of actomyosin bridge formation. We describe the stochastic binding/unbinding dynamics as the percolation process, wherein crosslinking sites are occupied by crosslinkers with a probability of 1/2 in every turnover time step $\tau$. If all sites (the total number $N$) are occupied by crosslinkers, an actomyosin bridge is formed between the cluster and the droplet boundary, and the cluster is transported toward the droplet boundary via actomyosin bridge contraction. The mean percolation time $\tau_p$ is defined as the number of time steps necessary to form the actomyosin bridge on average, and $k_{on}$ and $k_{off}$ indicate the binding and unbinding rates of crosslinkers, respectively. **c** An active gel model of periodic actomyosin wave formation. The wave period is determined by the sum of three sequential processes: (i) actin network formation by F-actin growth, (ii) stress generation by myosin binding, and (iii) the ring starts to contract toward droplet center.

$D_c = 85\,\mu m$ (Fig. 2k). The obtained value of $\tau$ is similar to the literature value $\tau = 0.54$ s, given that $k_{off} = 0.66\,s^{-1}$, $k_{on} = 1.2 \times 10^6\,M^{-1}\,s^{-1}$[36], and the concentration of actin and $\alpha$-actinin in *Xenopus* egg extracts are ~20 μM and $C_0 \sim 1$ μM, respectively[37]. Moreover, the obtained F-actin length ($L = 6.1$ μm) is also nearly consistent with the experimental value (Supplementary Fig. 12); the top 5% average of the F-actin length polymerized in the extracts gives 5.7 μm. Here we use this value in the length distribution for comparison because the tug-of-war model predicts that long filaments predominantly contribute to the network percolation (Supplementary Note 3).

The period of actomyosin waves is robust to the addition of the actin crosslinker ($\alpha$-actinin) and regulatory proteins of F-actin length (gelsolin, mDia2) (Fig. 5), suggesting that they predominantly modulate the dynamics of bulk actomyosin networks. This motivates us to further test if these factors can control $D_c$ in simulations with a fixed wave period $T(D)$. The minimal model shows that the longer F-actin length $L$ and higher crosslinker concentration $C_0$ (shorter turnover time $\tau = 1/(k_{off} + C_0 k_{on})$)

increases the percolation probability and in turn makes $D_c$ larger (Fig. 7). These trends are consistent with the experimental results of biochemical perturbations (Fig. 5): Eq. (2) is fitted to the edge-positioned probabilities in the presence of extra $\alpha$-actinin (Fig. 5, top) by fixing $L = 6.1$ μm (same as the control experiment) and setting $\tau$ as the only fitting parameter, giving $\tau = 0.09$ s. Based on the model, a reduction in $\tau$ indicates an increase in the crosslinker concentration $C_0$, consistent with the experiment. In experiments on actin length modulation (Fig. 5, middle and bottom), Eq. (2) is fitted to the edge-positioned probabilities by fixing $\tau = 0.46$ s (same as the control experiment) and setting $L$ as the only fitting parameter, and obtained $L = 4.3$ μm and $L = 10$ μm for gelsolin and mDia2 treatment experiments, respectively. The typical maximum lengths of F-actin (top 5% average) in the extracts are 3.2 μm (gelsolin) and 6.0 μm (mDia2), respectively (Supplementary Fig. 12). Although the values obtained from the model fitting are not completely matched with the measured values, trends of the shift of $L$ is fully consistent with the experiments (cf. 5.7 μm for the control case).

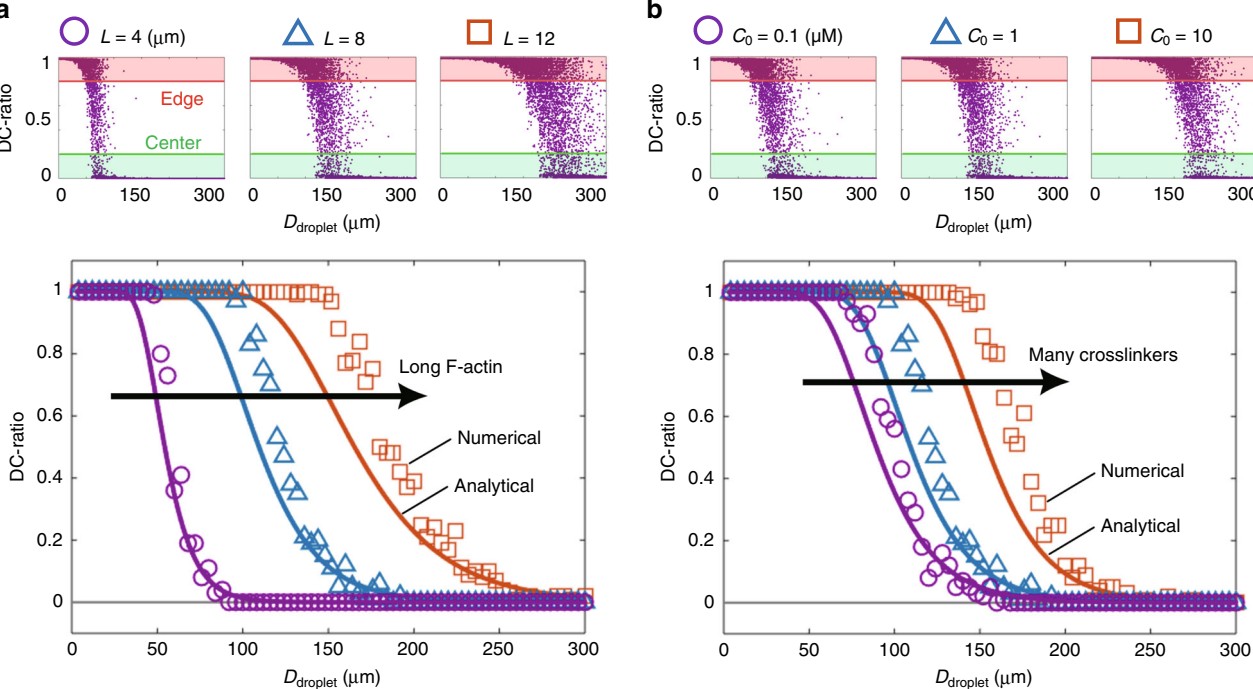

**Fig. 7 Numerical simulations and the analytical solution of the tug-of-war model reproduce the two-state cluster positioning.** Both numerical simulations and the analytical solution Eq. (2) of the tug-of-war model (Fig. 6) reproduce the results of perturbation experiments (Fig. 5). (Top) In numerical simulations, the time averaged DC-ratio $(1/T_0)\int_0^{T_0}[d(t)/R]\mathrm{d}t$ ($T_0 = 1800$ s) for each droplet was determined to statistically evaluate percolation dynamics, displayed as purple dots. (Bottom) Thereafter, the edge probability was determined from the DC-ratio, where a DC-ratio $> 0.8$ and $< 0.2$ was classified as the "edge" and the "center", respectively. **a** F-actin length $L$ was changed with the fixed crosslinker concentration ($C_0 = 1\,\mu$M). Purple circles, blue triangles, and orange squares represent numerical simulations with $L = 4\,\mu$m, $L = 8\,\mu$m, and $L = 12\,\mu$m, respectively. Solid curves are the corresponding analytical solutions. **b** Crosslinker concentration $C_0$ was changed with the fixed F-actin length ($L = 8\,\mu$m). Purple circles, blue triangles, and orange squares represent numerical simulations with $C_0 = 0.1\,\mu$M, $C_0 = 1\,\mu$M, and $C_0 = 10\,\mu$M, respectively. Solid curves are the corresponding analytical solutions. We defined different DC-ratio thresholds for the edge region from the experiments, because numerical simulations were performed for clusters without their finite volume, in which the centroid of the clusters is possible to reach the droplet boundary. For details, see Supplementary Note 4.

## Discussion

Previous works have reported a periodic gelation-contraction of actomyosin networks in millimeter-scale *Xenopus* egg extracts[21] and actomyosin networks reconstituted from purified proteins[11]. However, the actomyosin wave we observed in cell-sized droplets enveloped by natural phospholipids seems to have qualitatively different self-organization mechanisms. The clear ring-shaped profile (Fig. 1d) and the significant increase of F-actin density during its contraction (Supplementary Fig. 4), as well as quick response after the laser cut (Supplementary Fig. 5), indicate that strong contractile force is generated in the ring. Good agreement with the theoretical wave profiles also supports the ring-like nature of the wave (Fig. 1e, broken line, Supplementary Fig. 13). On the other hand, cluster formation inside cell-sized extract-in-oil droplets has been reported by several groups; confined extracts showed spontaneous F-actin flow and actin cluster formation[13], the temperature-dependent rupture and subsequent cluster formation[16], and the crosslinker concentration-dependent actin cluster formation[19]. So far, the extract-in-droplet system has been widely utilized for characterization of biological and rheological properties of actomyosin networks, yet the relationship between actomyosin waves and the cluster positioning has remained elusive. Here, we found the system size-dependent transition of the cluster positioning and we dissected its mechanism that the tug-of-war between actomyosin waves and actomyosin bridges control the positioning. The present study further pursuits the positioning in a cell-sized space that is playing important roles in

various biological functions, which could advance physical understanding and give insights into biological phenomena.

As for geometry sensing, for example, pole-to-pole oscillation of Min proteins in bacteria is known to guide formation of the symmetric division plane[38,39], and the cdk1 gradient guides direction of surface actomyosin waves toward the nucleus in starfish oocytes[40]. A distinct feature of actomyosin-driven geometry sensing revealed in this study is that it mechanically guides the positioning of an intracellular structure, rather than being biochemically guided by reaction-diffusion systems. To examine whether the mechanically guided centration mechanism can be extended into non-circular geometry, we encapsulated the extracts into non-circular chambers passivated with PEG. As a result, the actomyosin waves were robustly generated (Supplementary Movie 20), and the clusters mostly positioned near the geometric center even in the case of semicircular shaped chambers (Supplementary Fig. 15). Since the actomyosin bridge formation was suppressed by the surface passivation with PEG (Fig. 3b, top), this result implies that actomyosin waves can be used as a robust system to center nucleus-sized objects in various geometries. Interestingly, a periodic F-actin wave was observed in starfish and *Xenopus* embryos during cytokinesis[41]. Although relationships between the periodic F-actin wave, nuclear and spindle positioning in these organisms, and the periodic actomyosin wave observed in the present in vitro model remain unresolved, the actomyosin wave observed in living cells could be a possible regulator to find the geometric center.

In addition to the centripetal force generated by periodic actomyosin waves, the outward radial force generated by the actomyosin network percolation coexists in the same system. Our minimal model (Fig. 6) shows that the distinctive time-scales between two antagonistic forces are crucial to drive the two-state positioning: wave period $T$ has almost no size dependence, whereas the expected time for network percolation $\tau_p$ is strongly size-dependent. Since these two time-scales have different dependences on droplet size, the critical diameter for the symmetry breaking is determined by one unique intersection. The critical diameter $D_c$ can be analytically calculated from the equation $\tau_p = T$, with $\tau_p = 2^N \tau$ and $N \simeq R/L$, which yields

$$\frac{R_c}{L} = \log_2\left(\frac{T}{\tau}\right). \tag{3}$$

By using a typical wave period $T \simeq 46$ s, the typical maximum F-actin length $L = 5.7\,\mu m$ polymerized in the extracts (Supplementary Fig. 12), and the previously reported value of crosslinker turnover rate $\bar{\tau} = 0.54\,s^{36}$, the transition diameter is estimated as $\bar{D}_c = 2R_c \simeq 73\,\mu m$. This value is close to the experimentally obtained value $D_c = 85\,\mu m$ in the present study (Fig. 2k). Good agreement between the theory and experiments suggests that distinct system-size dependence in time-scales of periodic wave generation and network percolation is sufficient to direct the two-state cluster positioning.

Our results may provide physical insights into spindle and nuclear positioning. During oocyte maturation, the spindle migrates from the geometric center to beneath the oocyte surface, accompanied by an actin meshwork[7,8,42]. Recent studies have suggested that formin-mediated remodeling of bulk actin networks is involved in spindle migration[43]. Here, the present model indicates that these phenomena could be interpreted as increased percolation probability, wherein increased formin activity yields longer actin filaments in the cytoplasm to transmit contractile forces over a long distance (Fig. 5, bottom). The present results further suggest that regulations of actin crosslinkers might also be important because addition of $\alpha$-actinin increased the proportion of edge-positioned clusters (Fig. 5, top). Recently, attractive interaction of cytoplasmic aggregates with the boundary was observed in a similar extract-in-oil droplet system[44].

Since the nucleus always takes an asymmetric position in a migrating cell[1], the positioning mechanism found in this study might also help to understand the regulation of cell motility. Indeed, we found that the activation of Arp2/3 on the membrane placed the clusters to the droplet boundary even in large droplets (Fig. 3b, bottom), and this symmetry breaking initiated migration of droplets (Supplementary Movie 21). We expected that the edge-positioning of the cluster broke symmetry of the force balance in droplets, and actomyosin networks assembled beneath the droplet surface by Arp2/3 activation efficiently transmitted the inner force to the outer substrate, hence the droplets moved. During migration, the cluster positioned at the rear end of the droplet and actomyosin waves contracted toward it, resembling the rear-end positioning of the nucleus and retrograde actin flow observed in migrating cells. Further investigation of these interesting phenomena is remained as a future work.

Collectively, we demonstrated that the positioning symmetry of an intracellular structure is controllable by stoichiometric parameters of the actin cytoskeleton, providing a potential mechanism of actomyosin-dependent regulation of intracellular positioning in living cells. By taking the advantage of the in vitro system, we varied the droplet size over one order of magnitude to evaluate to what extent the surface to volume ratio of a confined space affects the cytoskeletal dynamics. This strategy enabled us to dissect two distinct characteristic time-scales coexisting in the same system, by which we identified that different self-

organization mechanisms between centripetal periodic actomyosin waves and radial network percolation are essential for the two-state positioning. Further investigation by using simplified in vitro models with the combination of in vivo systems will bring us better understandings of the regulatory mechanisms of biological functions orchestrated by the active cytoskeleton.

## Methods

**Preparation of *Xenopus* egg extracts.** Actin-intact cytoplasmic extracts were prepared from *Xenopus laevis* eggs[21]. Immediately thereafter, 1/1000 volume of protease inhibitors (10 mg mL$^{-1}$ leupeptin, 10 mg mL$^{-1}$ pepstatin A, 10 mg mL$^{-1}$ chymostatin dissolved in dimethyl sulfoxide (DMSO)), 1/20 volume of 2 M sucrose, and 1/20 volume of energy mix (150 mM creatine phosphate, 20 mM ATP, 20 mM MgCl$_2$) were added. The extracts were divided into 20 $\mu$L aliquots, snap-frozen in liquid nitrogen, and stored at $-80\,°C$. The interphase extracts were prepared by treating the metaphase extracts with 0.4 mM CaCl$_2$, followed by 30 min incubation at 20 °C[21].

**Preparation of recombinant proteins.** The VCA domain of mouse WASP cDNA (384-501 aa) was cloned into pCold-I vector (Takara), expressed in *E. coli* (Rosetta (DE3), Merck Millipore) at 15 °C for overnight in the presence of 1 mM IPTG. The ×6 histidine-tagged mutant was purified over a Ni Sepharose 6 Fast Flow column (GE healthcare), followed by dialysis against A50 buffer (50 mM HEPES–KOH pH 7.6, 50 mM KCl, 5 mM MgCl$_2$, 1 mM EGTA) containing 1 mM DTT at 4 °C.

Recombinant human $\alpha$-actinin I (×6 histidine-tagged) was prepared in accordance with our previous study[17], except for digestion of the histidine-tag by PreScission protease. Purified $\alpha$-actinin was dialyzed overnight against A150 buffer (50 mM HEPES–KOH pH 7.6, 150 mM KCl, 5 mM MgCl$_2$, 1 mM EGTA) containing 1 mM 2-mercaptoethanol at 2 °C.

The severing activity of native gelsolin is regulated by Ca$^{2+}$[45]. In this study, we constructed constitutively active form of gelsolin by deleting 23 amino acid residues from its C-terminus[45,46]. Mouse gelsolin cDNA (1-779 aa) was cloned into pCold-I vector (Takara), expressed in *E. coli* (Rosetta(DE3), Merck Millipore) at 15 °C for 15 h without IPTG[47]. The gelsolin mutant (×6 histidine-tagged) was purified over a Ni Sepharose 6 Fast Flow column (GE healthcare), followed by dialysis against A50 buffer containing 1 mM DTT at 4 °C.

The actin polymerization activity of native formins is regulated by the binding of Rho GTPases[48]. In this study, we constructed constitutively active form of formin by deleting the auto-inhibitory domain[49–51]. FH1 and FH2 domains of mouse mDia2 cDNA (521-1020 aa) were cloned into pGEX-6P vector (GE Healthcare), expressed in *E. coli* (Rosetta 2(DE3), Merck Millipore) at 16 °C for 2.5 h in the presence of 1 mM IPTG. The GST-tagged mDia2 mutant was purified over a Glutathione Sepharose High Performance column (GSTrap HP, GE Healthcare), followed by dialysis against A50 buffer containing 1 mM DTT at 4 °C.

Protein concentrations were determined using the Protein Assay Kit (500-0006, Bio-Rad), and using molecular weights of 15,200 Da for his-tagged VCA, 105,300 Da for his-tagged $\alpha$-actinin I, 80,100 Da for his-tagged gelsolin mutant, and 83,600 Da for GST-tagged mDia2 mutant. Proteins were snap-frozen in liquid nitrogen and stored at $-80\,°C$.

**Preparation of PDMS-coated glass slides.** Glass slides were coated with a silicone elastomer (polydimethyl siloxane (PDMS); Sylgard 184, Dow Corning). Uncured PDMS mixed with curing agent was poured on the glass slides, then PDMS was spread using a spin-coater (1000 rpm, 30 s). PDMS-coated glass slides were cured for 1 h at 75 °C. PDMS-coated coverslips (18 × 18 mm$^2$) were prepared by cutting the PDMS-coated glasses with a glass cutter.

**Encapsulation of extracts in droplets.** First, an aliquot of the extracts (20 $\mu$L) frozen in $-80\,°C$ was thawed and then incubated on ice for 1 h. Next, 0.2 $\mu$L of 100 $\mu$M TMR-LifeAct dissolved in DMSO (f. 1 $\mu$M) and 0.2 $\mu$L of 10 mM nocodazole dissolved in DMSO (f. 100 $\mu$M) were added to the extracts. At this point, the actin-associated proteins, actomyosin inhibitors, lipid-binding fluorescent marker R18 (Octadecyl Rhodamine B Chloride), and recombinant proteins were also added at a maximal volume of 0.2 $\mu$L. Then, 3–4 $\mu$L of the extracts was injected into 80 $\mu$L of lipid–oil mixture (1 mM L-$\alpha$-phosphatidyl choline from egg yolk (27554-01, Nacalai Tesque) in mineral oil (M5904, Sigma-Aldrich), which had been incubated on ice for more than 5 min. Alternatively, we also used 0.5% (w/v) PEG30-DPHS (Cithrol-SO-(JP), Croda) in mineral oil, or 10% DGS-NTA(Ni) (790404, Avanti Polar Lipids) in 1 mM of L-$\alpha$-phosphatidyl choline from egg yolk in mineral oil. By tapping the sample tube with a finger, we obtained various sizes of extract-in-oil droplets ranging from ~10 to 350 $\mu$m in diameter. Immediately after encapsulation, 10–50 $\mu$L of the emulsion was placed on a PDMS-coated glass slide and gently covered with a PDMS-coated coverslip on top. The chamber was sealed with epoxy putty or Valap to avoid fluid drift. The chamber height was controlled by the spacer thickness. Droplets with an aspect ratio of the chamber height $h$ to the droplet diameter $D_{droplet}$ between 0.3 and 0.6 were analyzed (16–330 $\mu$m in diameter). All procedures for preparation of the observation chamber were performed on an aluminum heat block placed on crushed ice.

**Preparation of observation chambers for bulk experiments.** First, air-plasma treatment was carried out using a plasma cleaner (PDC-32, Harrick Plasma). On placing PDMS-coated glass in the plasma cleaner and under vacuum for 30 s, surface activation was performed for 1 min at 18 W. After cleaning, the flow chamber covered with a PDMS-coated coverslip with the dimensions of $10 \times 18 \times 0.1$ mm$^3$ was constructed, then 1 mg mL$^{-1}$ of polylysine PEG conjugate (PEG-PLL) (PG2K-PLY, Nanocs) dissolved in 10 mM HEPES–NaOH pH 8.5 was perfused into the chamber. The positively charged lysine group electrostatically attaches onto the negatively plasma-activated PDMS surface, thereby PEG chains extending off the surface prevent protein adsorption onto the surface[52]. After the 30-min incubation in a 4 ℃ fridge, the PEG-PLL solution was air-dried immediately before use. Then, *Xenopus* egg extracts were placed on the PEG-PLL-coated glass and immediately covered with a PEG-PLL-coated coverslip, sealed by epoxy putty to prevent fluid drift (Supplementary Fig. 2b). Alternatively, a cut PCR tube was mounted on the PEG-PLL-coated glass and the rim of the tube was sealed by epoxy putty. Then, *Xenopus* egg extracts were gently placed inside the tube (Supplementary Fig. 2c).

**Encapsulation of the extracts in non-circular chambers.** Chromium masks (MITANI Micronics) were used to print patterns on SU-8 3025 photoresist (MicroChem) spin-coated on a silicon wafer, by using a mask-aligner (MA-100, MIKASA). The molds of PDMS microwells were constructed on the surface of silicon wafers. The surface of SU-8 patterns was smoothed by coating with a fluorinated coating agent CYTOP (Asahi Glass). PDMS elastomer was poured on the mold and coated at 300 rpm for 20 s, then cured at 70 ℃ for 1 h. The patterned PDMS sheet was gently placed on to a plasma-treated glass slide, heated for 1 h at 70 ℃ to make tight bonding between the PDMS sheet and the glass. Afterwards, a flow chamber was constructed on top of the PDMS sheet with a coverslip and 300 μm-thick double-sided tape (Supplementary Fig. 15). PEG-PLL solution was supplemented to the chamber and incubated for 3 h in a fridge at 4 ℃, which increased its hydrophilicity and to avoid non-specific adhesion of proteins. The PEG-PLL solution was dried by air duster. We then injected the extracts into the flow chamber. Subsequently, mineral oil with 0.5% PEG30-DPHS (w/v) was injected to the chamber to cover the micro-patterns with the monolayer of PEG30-DPHS. Finally, the flow chamber was sealed by epoxy glue to stop drift flow. All process was performed on ice to avoid acto-myosin contraction. Depth of the microwells was measured by a laser scanning surface profiler (LT-9000, Keyence) and it was about 200 μm.

**Microscopy.** Time-lapse images were acquired every 3 s using an epi-fluorescence microscope (IX73, Olympus) equipped with ×20 objective lens (TU Plan ELWD ×20/0.40, Nikon) for a chamber 100 μm high, or ×50 objective lens (TU Plan ELWD ×50/0.60, Nikon) for a chamber 60 μm high, or less, a cooled CMOS camera (Neo5.5, Andor Technology), and a stable excitation light source (XLED1, Lumen Dynamics). Confocal images were acquired using an inverted microscope (IX73, Olympus) equipped with ×40 objective lens (UPlanFL ×40/1.30 Oil, Olympus), a confocal scanner unit (CSU-X1, Yokogawa), 561 nm laser (50 mW; OBIS, Coherent), and EM-CCD camera (iXon3, Andor Technology). The laser ablation experiments were performed by using a UV pulse laser (Explorer One 349 nm, Spectra-Physics), equipped to the spinning-disk confocal microscope. For all microscopic examinations, the sample temperature was maintained at 20 ± 1 ℃, using a home-made heat block connected to a water bath circulator.

**Length measurements of actin filaments.** The extracts were incubated at 20 ℃ for 1 h. Then, the extracts were diluted 20× by a high salt buffer (20 mM Tris–HCl pH 7.4, 0.6 M KCl, 1 mM DTT) containing 0.5 μM rhodamine phalloidin, in order to dissociate actin crosslinkers from actin bundles and stabilize the filaments[53]. The solution was further diluted by the high salt buffer to lower the filament density and perfused into NEM–HMM-coated flowchamber to fix actin filaments on the glass surface, then the images were taken by an epi-fluorescence microscopy equipped with ×100 objective lens (PlanApo ×100/1.40 Oil, Olympus) and EM-CCD camera (iXon3, Andor Technology). The filament length was measured by a custom code written in MATLAB. First, the fluorescence images of F-actin were binarized. Then, only skinny and unbranched filaments (eccentricity > 0.9) and those with reasonable sizes (area < 1300 pixels) were selected. Finally, the filaments were converted into one-pixel-wide sticks, by which the area of sticks was equal to the filament length.

**Image analysis.** Quantitative image analysis was performed using a custom code written in MATLAB. In particular, the center of mass of each droplet and cluster was detected through binarization of bright field images. To determine the radius $R_{droplet}$ of droplets, the area of each droplet was extracted and then assumed to be equal to $\pi R_{droplet}^2$. The radius of clusters was derived in the same manner. Quantification of the contraction velocity of actomyosin waves and determination of the wave period were performed by producing kymographs of fluorescence images along the droplet diameter. Notably, actomyosin waves displayed a biphasic velocity profile; the contraction velocity was almost constant during the initial phase and decreased rapidly when the wave reached near the cluster. The initial contraction velocity was determined from the angle of the kymograph (Fig. 1e, Supplementary Fig. 6a). The period was determined from the distance between two waves in the kymograph. Then, they were averaged over

three successive waves and periods. To visualize the spatial velocity profile of the actomyosin wave, we used a public domain particle imaging velocimetry (PIV) program implemented as a Fiji/ImageJ plugin. Organelles transported by the periodic actomyosin wave observed in bright-field images were sequentially analyzed to obtain the flow field (Fig. 1g).

**Statistical analysis.** Statistical analysis was carried out using Microsoft Excel and MATLAB. The $p$-values were determined by Kolmogorov–Smirnov test.

**Theoretical models and numerical simulations.** The actomyosin periodic wave was modeled based on the active gel theory[54], and its contractile period and velocity were calculated analytically. The tug-of-war hypothesis was modeled by invoking the percolation theory. The details of these theoretical models are given in Supplementary Notes 1–3, respectively. Furthermore, we tested the derived formula (Eq. (2)) by stochastic numerical simulations, as elaborated in Supplementary Note 4.

**Estimation of the transition droplet diameter.** An analytical expression of the tug-of-war model (Eq. (2)) was fitted to experimentally obtained edge-probabilities by non-linear least-square regressions. From the fitting curve, the transition droplet diameter $D_c$ is determined as the droplet diameter at which the edge-probability becomes $p(D_c) = 0.5$.

**Reporting summary.** Further information on research design is available in the Nature Research Reporting Summary linked to this article.

## Data availability

Data supporting the findings of this manuscript are available from the corresponding author upon reasonable request. A reporting summary for this Article is available as a Supplementary Information file. The source data underlying Figs. 1–7 and Supplementary Figs. 1–15 are provided as a Source Data file. Source data are provided with this paper.

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

## Acknowledgements

We thank Z. Izri for fruitful discussion. This work was partly supported by Grants-in-Aid for Challenging Exploratory Research (grant no. 15K14497 to M.M.), Young Scientists (A) (grant no. 16H06165 to M.M.), Young Scientists (B) (grant no. 16K17777 to T.H.), Scientific Research (B) (grant no. 17KT0025 to Y.T.M., grant no. 16KT0077 to M.M.), Scientific Research (S) (grant no. 22227005 to S.I.), Scientific Research on Innovative Areas "Molecular Engines" (grant no. 18H05427 to Y.T.M., grant no. 19H05393 to M.M.), and JSPS Fellows (grant no. JP19J20035 to R.S.) from the Ministry of Education, Culture, Sports, Science, and Technology, Japan; Human Frontier Science Program Research Grant (grant no. RGP0037/2015 to Y.T.M.); Waseda University Grant for Special Research Projects (grant no. 2015S-086 to M.M.); Kishimoto Foundation Research Grant from the Senri Life Science Foundation (to M.M.); and The Hakubi project of Kyoto University (to M.M.).

## Author contributions

Y.T.M., R.S., and M.M. designed experiments. M.T. and K.S. developed the actin-intact egg extract system. M.M. prepared proteins. R.S. and M.T. performed experiments. R.S. and M.T. analyzed the data. T.H. constructed the theoretical model. R.S. performed numerical simulations. All the authors discussed the results. R.S., S.I., Y.T.M., and M.M. wrote the manuscript.

## Competing interests

The authors declare no competing interests.
