## [Peer Review File · Nature Communications]

Reports from round 1:

Rev#1

The authors have done an excellent job addressing the reviewer concerns, including addition of significant new experiments, all of which support their model. All critiques have been addressed, this is a lovely paper.

Rev#2

The authors have sufficiently addressed referee concerns. The manuscript is improved and appropriate for publication in Nature Communications

Rev#3

The authors went a long way to answer my previous and other reviewer comments. Overall the manuscript has improved with the new experiments and the qualitative understanding of the process seems to be supported by the data. I only have two concerns left.

One minor is that the authors provide new analysis to distinguish between bulk contraction vs ring contraction. For that they refer to the new supplementary figure 4 that shows that the intensity of the ring goes up as it contracts whereas the bulk intensity remain constant. This would indeed suggest that the contraction is happening in the ring. However, this seems to contradict their exact same measurements in supplementary figure 2 :“(f) Fluorescence intensity along the each broken line in e, which also corresponds to the intensity at the arrows with the same color in d. Both the F-actin densities on the rim and inside of the wave increased during the contraction. The thickness of the spacer was 100 μm . Scale bars, 1 mm. ” Could the authors clarify the discrepancy? Does it depend on day to day or example taken?

My main concern is still related to the theory:

For their theory the authors state: “We have proposed that the off-centering is mediated by the percolated actomyosin bridge”, and that the bridge emerges from “low number of remained F-actin and newly assembled F-actin stochastically form bridges between the cluster and the boundary.” The authors are assuming the bridge is pre-formed and just calculate the probability that cross linkers connect the filaments that form the bridge. But, why don't we see the bridge than initially after the wave? From their images it looks that actin is actually polymerized as the bridge is form (See figure 4c. After laser ablation a new bridge is formed but no wave has formed before it). Since the authors only consider the top 5% of actin lengths, shouldn't this have a dramatic effect on time scales too? Meaning that it may be very difficult to actually have a bridge made of actin filaments longer than the 5% population as opposed of having a bridge made of short actin filaments. Of course that time scale to percolate a bridge of shorter filaments is longer, but it may be compensated by the time scales (or probability required to create enough long ($\sim 6 \mu\text{m}$) filaments. Also, how can the authors really justify to use 6 μm as length based on their measured length distributions? These distributions clearly show an average of $\sim 1 \mu\text{m}$, and almost no filaments larger than 5 μm . I think it is necessary that the authors clarify this point, comment on the origin of actin in these bridges, what would be the time scales associated to having only the 5% population of actin, and recalculate their predictions based on the average actin length they measure.

The answer the authors give to use $\sim 6 \mu\text{m}$ is flawed. They argue “In the proposed model, the probability that a bridge is formed by the length L of F-actin in a droplet with radius R is written as $P=(1/2)^{R/L}$. If we compare the probability with different actin length, $L=10 \mu\text{m}$ and $L=5 \mu\text{m}$, in a droplet with radius $R=50 \mu\text{m}$, we can estimate $P=(1/2)^5$ and $P=(1/2)^{10}$ respectively. Thus, only 2 times change in F-actin length decreases the bridge formation probability ~ 30 times.”. The authors are missing the probability of actually getting actin filaments of that particular length. With their measured length distribution, this can also change by a decade from ~ 1 to $\sim 5 \mu\text{m}$ so it could counteract their estimated probability, so the authors need to take this into account in their argument.

The authors distinguish gelation vs percolation to explain the difference of timescales, which they

argue leads to a faster gelation time. However, my understanding is that gelation is a connectivity transition, and that previous research has used percolation as a theory to explain the gelation process. In both cases, they are network-connectivity transitions. Gelation further implies the emergence of material properties such as a finite young modulus, whereas percolation theory makes no statement (or can't explain) about it and it is just restricted to the emergence of a system size cluster. Thus, I don't really understand what the authors imply by "Since percolation is possible even without gelation, it is natural to assume that the bridge formation does not have to rely on the gelation process.", and why would we expect that the percolation would take longer? Could the authors explain what the explicit differences between percolation and gelation they are implying?

Response Letter:

Tug-of-war between actomyosin-driven antagonistic forces determines the positioning symmetry in cell-sized confinement

R. Sakamoto, M. Tanabe, T. Hiraiwa, K. Suzuki, S. Ishiwata, Y. T. Maeda, and M. Miyazaki

We greatly appreciate all the reviewers for careful reading of our manuscripts and giving us fruitful comments. We provide our response to each comment from Reviewer #3 below.

Response to Reviewer #3

The authors went a long way to answer my previous and other reviewer comments. Overall the manuscript has improved with the new experiments and the qualitative understanding of the process seems to be supported by the data. I only have two concerns left.

One minor is that the authors provide new analysis to distinguish between bulk contraction vs ring contraction. For that they refer to the new supplementary figure 4 that shows that the intensity of the ring goes up as it contracts whereas the bulk intensity remain constant. This would indeed suggest that the contraction is happening in the ring. However, this seems to contradict their exact same measurements in supplementary figure 2 : “(f) Fluorescence intensity along the each broken line in e, which also corresponds to the intensity at the arrows with the same color in d. Both the F-actin densities on the rim and inside of the wave increased during the contraction. The thickness of the spacer was 100 μm . Scale bars, 1 mm.” Could the authors clarify the discrepancy? Does it depend on day to day or example taken?

We are thankful to the reviewer for raising this point. The droplet size makes the difference. While the large droplet with the radius $R \sim 1.5 \text{ mm}$ exhibits global contraction of the bulk actomyosin network (**Supplementary Figure 2**), the small droplet with the radius $R \sim 150 \mu\text{m}$ shows ring-like actomyosin waves (**Supplementary Figure 4**). This point has been mentioned in the main text (**page 5-6, lines 103-124**).

The origin of the difference could be explained by the different size scaling between the inward and outward material transport. The active network contraction transports materials to the droplet center. Since the contraction velocity v is almost proportional to the droplet radius R (**Figure 1i**), the timescale of the inward material transport is almost independent of the droplet size ($\tau_{\text{in}} = R/v \sim \text{const.}$). On the other hand, materials accumulated at the cluster

Response Letter:

Tug-of-war between actomyosin-driven antagonistic forces determines the positioning symmetry in cell-sized confinement

R. Sakamoto, M. Tanabe, T. Hiraiwa, K. Suzuki, S. Ishiwata, Y. T. Maeda, and M. Miyazaki

diffuse to the periphery of the droplet to sustain the periodic waves. Because the timescale of diffusion τ_{out} scales quadratically with the droplet size as $\tau_{\text{out}} \sim R^2/D$ (D : the diffusion constant), it takes much longer time for the materials to reach near the periphery in larger droplets. Thus, we suppose that the amount of the constituents of the wave could become insufficient to make the clear contrast of the F-actin fluorescence intensity between the rim and inside of the wave in large droplets. Although we currently do not know the exact mechanism, the main focus of the present study is the positioning mechanism in a cell-sized confined space. We think that dissecting the origin of the difference is the key to fully understand the physical mechanism of the periodic actomyosin wave generation. A detailed investigation would be our future work.

My main concern is still related to the theory: For their theory the authors state: “We have proposed that the off-centering is mediated by the percolated actomyosin bridge”, and that the bridge emerges from “low number of remained F-actin and newly assembled F-actin stochastically form bridges between the cluster and the boundary.” The authors are assuming the bridge is pre-formed and just calculate the probability that cross linkers connect the filaments that form the bridge. But, why don't we see the bridge than initially after the wave? From their images it looks that actin is actually polymerized as the bridge is form (See figure 4c. After laser ablation a new bridge is formed but no wave has formed before it). Since the authors only consider the top 5% of actin lengths, shouldn't this have a dramatic effect on time scales too? Meaning that it may be very difficult to actually have a bridge made of actin filaments longer than the 5% population as opposed of having a bridge made of short actin filaments. Of course that time scale to percolate a bridge of shorter filaments is longer, but it may be compensated by the time scales (or probability) required to create enough long (~6 μm) filaments.

We greatly appreciate the reviewer's comments to improve the quality of the paper. First of all, to avoid possible confusions, we have to clarify how fast the time scale of actin polymerization is compared to the other time scales in our system. Accordingly, we estimate the polymerization time to reach the filaments enough long (~6 μm). Given that an actin filament elongates by 2.5 nm with the addition of an actin monomer with the diameter of $d \sim 5$ nm [1], and by using the

Response Letter:

Tug-of-war between actomyosin-driven antagonistic forces determines the positioning symmetry in cell-sized confinement

R. Sakamoto, M. Tanabe, T. Hiraiwa, K. Suzuki, S. Ishiwata, Y. T. Maeda, and M. Miyazaki

concentration of actin in the *Xenopus* egg extracts ($\sim 20 \mu\text{M}$) [2] and the polymerization rates of filaments ranging from ~ 300 molecules/s (*in vitro*) [3] to ~ 136 molecules/s [4] and $\sim 2.0 \mu\text{m/s}$ [5] (*in vivo*), $6 \mu\text{m}$ -long filaments will be formed in $3 \text{ s} \sim 18 \text{ s}$. Since the wave period is $\sim 46 \text{ s}$, we expect that the long filaments could exist from the early stage after the wave. In other words, the length distribution could reach a steady-state soon after the wave. Note that, because it would be reasonable to expect that not all the actin filaments are nucleated from single monomers but some of them are elongated from F-actin fragments with a certain length, the polymerization timescale could be shorter than our estimation.

In addition, because the short actin filaments are more abundant in the extracts, the enlarged background noise in fluorescence could easily mask the weak signal of the bridge of long F-actin just after the wave generation. Therefore, we believe that this length distribution itself makes it difficult to visualize the bridge of long F-actin.

Also, how can the authors really justify to use $6 \mu\text{m}$ as length based on their measured length distributions? These distributions clearly show an average of $\sim 1 \mu\text{m}$, and almost no filaments larger than $5 \mu\text{m}$. I think it is necessary that the authors clarify this point, comment on the origin of actin in these bridges, what would be the time scales associated to having only the 5% population of actin, and recalculate their predictions based on the average actin length they measure. The answer the authors give to use $\sim 6 \mu\text{m}$ is flawed. They argue "In the proposed model, the probability that a bridge is formed by the length L of F-actin in a droplet with radius R is written as $P=(1/2)^{R/L}$. If we compare the probability with different actin length, $L=10 \mu\text{m}$ and $L=5 \mu\text{m}$, in a droplet with radius $=50 \mu\text{m}$, we can estimate $P=(1/2)^5$ and $P=(1/2)^{10}$ respectively. Thus, only 2 times change in F-actin length decreases the bridge formation probability ~ 30 times.". The authors are missing the probability of actually getting actin filaments of that particular length. With their measured length distribution, this can also change by a decade from ~ 1 to $\sim 5 \mu\text{m}$ so it could counteract their estimated probability, so the authors need to take this into account in their argument.

We greatly appreciate the reviewer's careful consideration about the assumptions of our model. The reviewer suggests that one may need to use the

Response Letter:

Tug-of-war between actomyosin-driven antagonistic forces determines the positioning symmetry in cell-sized confinement

R. Sakamoto, M. Tanabe, T. Hiraiwa, K. Suzuki, S. Ishiwata, Y. T. Maeda, and M. Miyazaki

average actin length for the quantitative estimation rather than the typical maximum length, which is the assumption of our model. However, there is the proper reason by which we have to use not the average but the typical maximum length of F-actin in our estimation. Firstly, we will explain the reason qualitatively. After that, we will put the quantitative evaluation of the edge-positioning probability by taking into account the length distribution of F-actin to support the qualitative explanation.

Qualitatively, there are two reasons why we use the typical maximum length L for $N = R/L$. (i) Firstly, short filaments can hardly line up across the center to the periphery of the droplet over the radius R by chance. For example, please imagine the extreme situations that there are one filament of length $L=R$ with the probability distribution of 0.01, and five filaments of length $L=R/5$ with the probability distribution of 0.1. The probability that the five filaments of length $L=R/5$ line up across R by chance, $(0.1)^5$, is much smaller than the probability that the filament with length $L=R$ connects the same distance R by chance, 0.01. (ii) In addition, the typical percolation probability via crosslinker binding decreases exponentially for decreased actin length L , as we discussed in the previous response letter to the reviewer. For example, given that the filament of length $L=R$ and $L=R/5$, we can estimate the percolation probability as $(1/2)^{R/L} = (1/2)^1$ and $(1/2)^5$, respectively. Together, even if we consider the effect of the length distribution of F-actin, the maximally long filaments may predominantly contribute to the percolation. We have added this discussion in **Supplementary Note 3 (page 24, lines 284-297)** to provide the rationale for the use of a typical maximum length. Corresponding to this change, we have also added the sentence “Here we use this value in the length distribution for comparison because the tug-of-war model predicts that long filaments predominantly contributes to the network percolation (Supplementary Note 3)” in the Main text **(page 14, lines 312-314)**.

Quantitatively, here we calculate the edge-positioning probability with taking into account the length distribution of F-actin. If we know the length distribution $p(l)$, the probability that the filaments with length l align along the droplet radius R is written as $P_b(l) = p(l)^{R/l} / \int_0^{l_{max}} p(l)^{R/l} dl$, where l_{max} is the longest length of F-actin in the distribution. Next, as we discussed in the main text, the probability that the all $N(=R/l)$ binding sites are not occupied by crosslinkers is written as

Response Letter:

Tug-of-war between actomyosin-driven antagonistic forces determines the positioning symmetry in cell-sized confinement

R. Sakamoto, M. Tanabe, T. Hiraiwa, K. Suzuki, S. Ishiwata, Y. T. Maeda, and M. Miyazaki

$g(l) \equiv 1 - (1/2)^{R/l}$. Therefore, the probability that there is no bridge consisting of any F-actin length between $l_{min} < l < l_{max}$ is written as

$$g(l_{min}) * g(l_1) * g(l_2) * \dots * g(l_{max}) = \exp \left[\sum_{l_{min}}^{l_{max}} \log \{g(l)\} \right], \quad (R1)$$

By taking into account the contribution of the length distribution of F-actin, Eq.(R1) can be rewritten as $\exp \left[\int_{l_{min}}^{l_{max}} \log\{g(l)\} P_b(l) dl \right]$. Finally, the probability that the bridge formation occurs at least once during period T is given by

$$p(R) = 1 - \left(\exp \left[\int_{l_{min}}^{l_{max}} \log\{g(l)\} P_b(l) dl \right] \right)^{T/\tau}, \quad (R2)$$

which corresponds to the edge-positioning probability Eq.(2) in the main text (**page 13, line 296**).

By using Eq.(R2), we compare the following two cases: the edge-positioning probability calculated by using (i) 100% distribution, or (ii) only top 5% distribution of actin filaments (**Response Figure 1**). Firstly, for simplicity, we assume that the length distribution of F-actin is given by $p(l) = (1/L_A) * \exp(-l/L_A)$, where the mean value of the control experiment ($\sim 1.8 \mu\text{m}$, **Fig. S12a**) is used for L_A . Next, l_{max} is adjusted to match the curve (R2) to the original curve of the control experiment (**Fig. 2k**) with using the same values for T and τ used in the main text (**page 13, line 296**). The fitting gives $l_{max} = 7.0 \mu\text{m}$. Then, we change only l_{min} value and compare the following two cases: (i) using 100% distribution, (ii) using only top 5% distribution. The curves calculated by using 100% distribution (green line; $l_{max} = 7.0 \mu\text{m}$, $l_{min} = 0 \mu\text{m}$) and by using top 5% distribution (orange line; $l_{max} = 7.0 \mu\text{m}$, $l_{min} = -L_A * \log(0.05 + (1-0.05) * \exp(-l_{max}/L_A)) = 4.8 \mu\text{m}$) are mostly overlapped. Thus, we can conclude that it is reasonable to assume that the top 5% of filaments predominantly contributes to the bridge formation.

Since we found that the top 5% predominantly contributes to the bridge formation, we can approximate Eq.(R2) by assuming that only the filaments with a typical maximum length L participate in the bridge formation. Under this approximation, we can rewrite $l \sim L$ and $P_b \sim 1/(l_{max} - l_{min})$ in Eq.(R2), which finally leads to Eq.(2) in the main text (**page 13, line 296**).

Response Letter:

Tug-of-war between actomyosin-driven antagonistic forces determines the positioning symmetry in cell-sized confinement

R. Sakamoto, M. Tanabe, T. Hiraiwa, K. Suzuki, S. Ishiwata, Y. T. Maeda, and M. Miyazaki

Response Figure 1 | The edge-positioning probabilities calculated by using 100% distribution or only top 5% distribution of actin filaments.

The authors distinguish gelation vs percolation to explain the difference of timescales, which they argue leads to a faster gelation time. However, my understanding is that gelation is a connectivity transition, and that previous research has used percolation as a theory to explain the gelation process. In both cases, they are network-connectivity transitions. Gelation further implies the emergence of material properties such as a finite young modulus, whereas percolation theory makes no statement (or can't explain) about it and it is just restricted to the emergence of a system size cluster. Thus, I don't really understand what the authors imply by "Since percolation is possible even without gelation, it is natural to assume that the bridge formation does not have to rely on the gelation process.", and Why would we expect that the percolation would take longer? Could the authors explain what the explicit differences between percolation and gelation they are implying?

We are grateful to the reviewer for raising this point to make the concept of the study clearer. We agree with the reviewer's comment and realize that the term "gelation" used in the present study will mislead general readers including theoretical physicists and material scientists. Because we have not intended to use "gelation" in terms of rheological properties which we have not quantified in the present study, we replaced "gelation" in the main text with "growth" (**page 12, line 270**) or "maturation" (**page 13, line 281**), and the corresponding phrases in the Supplementary Notes.

Response Letter:

Tug-of-war between actomyosin-driven antagonistic forces determines the positioning symmetry in cell-sized confinement

R. Sakamoto, M. Tanabe, T. Hiraiwa, K. Suzuki, S. Ishiwata, Y. T. Maeda, and M. Miyazaki

In the previous manuscript, we referred to “gelation” as the process that actin filaments are actively polymerized and branched by actin-nucleation factors near the droplet boundary to form the ring-shaped actomyosin network. On the other hand, we referred to “percolation” as the bridge formation process by the stochastic binding/unbinding of crosslinkers between actin filaments existing in the bulk space. To summarize, we used “percolation” and “gelation” to distinguish two distinct organization processes of actomyosin networks occurred in different places in the droplet.

Because maturation of the ring-shaped actomyosin network locally proceeds via each actin-nucleation factor, we expect that the timescale for this process (referred to “gelation” in the previous manuscript) has a small system size dependence. In contrast, because bridge formation between the cluster and droplet boundary is achieved globally across the bulk space via simultaneous crosslinking of all the filaments aligned along the radial direction, we expect that the timescale for this process (referred to “percolation” in the previous and present manuscripts) exponentially increases as the droplet size increases. Considering these distinct organization processes and system-size dependences, it will be reasonable to expect that the bridge formation could take longer time than the maturation of the ring-shaped actomyosin network in large droplets.

References

1. Holmes, K. C., Popp, D., Gebhard, W. & Kabsch, W. Atomic model of the actin filament. *Nature* **347**, 44-49 (1990).
2. Wühr, M. *et al.* Deep proteomic of the *Xenopus laevis* egg using an mRNA-derived reference database. *Curr. Biol.* **24**, 1467-1475 (2014).
3. Pollard, T. D. Rate Constants for the Reactions of ATP- and ADP-Actin with the Ends of Actin Filaments. *J. Cell Biol.* **103**, 2747-2754 (1986).
4. Fritzsche, M., Erlenkämper, C., Moeendarbary, E., Charras, G. & Kruse, K. Actin kinetics shapes cortical network structure and mechanics. *Sci. Adv.* **2**, e1501337 (2016).
5. Higashida, C. *et al.* Actin Polymerization–Driven Molecular Movement of mDia1 in Living Cells. *Science* **303**, 2007-2010 (2004).